# Mitigating Bias in Locally Constrained Decoding via Tractable Proposals

**Meihua Dang** [1]  **Linxin Song** [2]  **Honghua Zhang** [3]  **Jieyu Zhao** [2]  **Guy Van den Broeck** [3]  **Stefano Ermon** [1]

## Abstract

Generations from large language models often fail to conform to desired constraints such as JSON schema. Existing locally constrained decoding (LCD) approaches enforce constraints by myopically masking out next tokens, resulting in biased sampling and degradation in performance. Recent work uses sequential Monte Carlo (SMC) methods to mitigate such biases, but designing effective proposal distributions or potential functions remains a key challenge. In this work, we propose a generic approach to construct proposals and potentials for SMC sampling from $p_{\text{lm}}(\cdot \mid \text{constraint})$. First, we show that constraints specified as finite automata can be tensorized for efficient execution on GPUs, which we use to construct *globally constrained decoding* (GCD) proposals. In addition, leveraging the fact that tensorized finite automata share the same *circuit structure* as hidden Markov models, we circuit-multiply them to obtain the *probabilistic GCD* (P-GCD) proposals encoding both logical and probabilistic information about the target distributions. We evaluate (P-)GCD on the tasks of function calling, keyword-based generation, and SQL generation. Experiments show that under the same SMC sampling setup, compared to LCD proposals, (P-)GCD converges faster to the target distribution with significantly fewer particles.

## 1. Introduction

Many applications require LLM outputs to satisfy strict logical or structural constraints. Examples include generating function calls conforming to a JSON schema (Patil et al., 2024; 2025), producing syntactically valid SQL queries (Lei

et al., 2025), and avoiding toxic content (Gehman et al., 2020). Despite substantial advances in instruction fine-tuning (Wei et al., 2022; Chung et al., 2024) and preference optimization (Ouyang et al., 2022; Rafailov et al., 2023), current LLMs remain unreliable at consistently enforcing such hard constraints (Lu et al., 2023).

Given a constraint $\mathcal{C}$, the simplest unbiased way to sample from the constrained distribution $p_{\text{lm}}(x_{1:n} \mid \mathcal{C})$ is *rejection sampling*: repeatedly draw from the unconstrained LM and reject those violating the constraint. However, it becomes impractical for complex constraints due to vanishingly low acceptance rates. A standard alternative is *locally constrained decoding (LCD)*, which masks out next tokens that immediately violate the constraint at each decoding step (Scholak et al., 2021; Dong et al., 2025). However, this local strategy (i) distorts the model's distribution (Park et al., 2024) and (ii) does not guarantee constraint satisfaction *within a finite token budget*, since the model may continue generating locally valid prefixes indefinitely without ever reaching a satisfying completion (Li et al., 2025).

To address the two limitations of LCD, we propose:

1. **Globally constrained decoding (GCD)**. Given constraints encoded as finite automata, we compile them into tensorized representations for efficient execution on GPUs. We use them to efficiently mask tokens that cannot lead to a valid completion within a token budget. This guarantees constraint satisfaction at termination.

2. **Probabilistic GCD (P-GCD)**. We first distill hidden Markov models to approximate the LM's distributions, following Zhang et al. (2023). Then we efficiently multiply them with the finite automata, leveraging circuit multiplication and tensor-network operations (Loconte et al., 2025). The multiplied distributions capture both hard constraint satisfaction and probabilistic information about future valid completions.

We compare (P-)GCD against LCD within the *sequential Monte Carlo* (SMC) sampling framework. Prior work applies SMC with LCD as the proposal distribution, mitigating its bias by re-weighting and re-sampling a batch of sequences (*particles*) (Loula et al., 2025; Lipkin et al., 2025). SMC is guaranteed to converge to the target distribution as the number of particles increases, but the convergence rate is bottlenecked by the quality of the proposal.

---

[1]Department of Computer Science, Stanford University [2]Department of Computer Science, University of Southern California [3]Department of Computer Science, University of California, Los Angeles. Correspondence to: Meihua Dang <mh-dang@cs.stanford.edu>.

*Proceedings of the 43rd International Conference on Machine Learning*, Seoul, South Korea. PMLR 306, 2026. Copyright 2026 by the author(s).

We evaluate (P-)GCD on three constrained generation benchmarks: xLAM, a function-calling dataset with outputs constrained to follow JSON or Python-like syntax; Common-Gen, a keyword-based generation task; and Spider, a text-to-SQL task where outputs must follow SQL syntax. When used as the proposal or potential distribution within the SMC framework, (P-)GCD outperforms LCD-based proposals in both generation quality and sampling efficiency, converging to the target distribution with significantly fewer particles.[1]

## 2. Problem Setup

**Notation.** We denote a sequence of $n$ tokens as $x_{1:n}$, with $x_t$ the token at position $t$ and $x_{<t}$ the prefix $x_1, \ldots, x_{t-1}$, and assume all sequences generated by language models are padded to length $n$. We use $p$ to denote a distribution defined by a model (e.g., $p_{\mathrm{lm}}$), $q$ to denote any modified distribution used for sampling, and $\mathcal{C}$ to denote any constraint.

**Constrained Generation.** The goal of this work is to sample sequences from an LLM that strictly satisfy a given logical constraint $\mathcal{C}$. Let $p_{\mathrm{lm}}(x_{1:n}) = \prod_{t=1}^{n} p_{\mathrm{lm}}(x_t \mid x_{<t})$ denote the LLM distribution over token sequences of length $n$. The target distribution is then

$$p_{\mathrm{lm}}(x_{1:n} \mid \mathcal{C}) \propto p_{\mathrm{lm}}(x_{1:n}) \cdot \mathbf{1}\{x_{1:n} \in \mathcal{X}_n\},$$

where $\mathbf{1}\{\cdot\}$ denotes the indicator function and $\mathcal{X}_n := \{ x_{1:n} : x_{1:n} \text{ satisfying } \mathcal{C} \}$.

**Rejection Sampling.** The simplest and unbiased approach to sample from the target distribution $p_{\mathrm{lm}}(x_{1:n} \mid \mathcal{C})$ is rejection sampling: repeatedly drawing $x_{1:n} \sim p_{\mathrm{lm}}$ until $x_{1:n} \in \mathcal{X}_n$. However, this is impractical for non-trivial constraints, where acceptance rates can be vanishingly low.

**Locally Constrained Decoding.** During autoregressive generation, one can often detect whether a candidate next token would immediately violate the constraint. For example, a valid JSON object must begin with an opening brace "{" or bracket "["; sampling anything else at the first step immediately violates the constraint. Locally constrained decoding (LCD) exploits this by *masking out* invalid tokens at each sampling step and renormalizing distribution over the tokens that still permit a valid completion from the current prefix. That is, it defines the proposal distribution:

$$q_{\mathrm{lcd}}(x_t \mid x_{<t}) \propto p_{\mathrm{lm}}(x_t \mid x_{<t}) \cdot \mathbf{1}\{x_{1:t} \in \mathcal{X}_{t,*}\}, \quad (1)$$

where $\mathcal{X}_{t,m} = \{ x_{1:t} : \exists\, x_{t+1:m},\ x_{1:m} \text{ satisfying } \mathcal{C} \}$ is the set of prefixes that can be extended to a length-$m$ sequence satisfying $\mathcal{C}$, and $\mathcal{X}_{t,*} = \bigcup_{m \geq t} \mathcal{X}_{t,m}$ allows extension to satisfying sequences of *any* length. In other words, LCD

rejects $x_t$ only when no valid sequence of any length begins with the prefix $x_{1:t}$ (Lipkin et al., 2025). However, this is a "loose" masking criterion: locally valid prefixes may only admit a valid completion beyond length $n$, so the final length-$n$ sequence may still violate the constraint $\mathcal{C}$. For example, when generating a JSON object, the decoder may repeatedly emit unmatched opening braces "{". Each step remains locally valid, but if generation stops at length $n$ before the braces are closed, the final sequence is invalid. There are a variety of open-source constrained decoding frameworks, such as Outlines (Willard & Louf, 2023) and XGrammar (Dong et al., 2025), and constrained decoding is widely deployed in production for proprietary models, such as the OpenAI API (Geng et al., 2025). To the best of our knowledge, all of the aforementioned systems perform LCD and do not guarantee constraint satisfaction *within a given token budget*. Our approach eliminates such failure cases (Section 3). Moreover, the greedy and myopic enforcement of LCD *deviates from* the target conditional distribution $p_{\mathrm{lm}}(x_{1:n} \mid \mathcal{C})$, introducing substantial sampling bias (Loula et al., 2025; Park et al., 2024); see Appendix A for an illustrative example.

**Sequential Monte Carlo Sampling.** To approximately sample from the target distribution $p_{\mathrm{lm}}(x_{1:n} \mid \mathcal{C})$ while minimizing the aforementioned biases, Loula et al. (2025) propose to use SMC with LCD as the proposal distribution, followed by importance reweighting to correct samples toward the target. Concretely, the SMC sampling algorithm maintains $k$ particles representing partial sequences, indexed by $i = 1, \ldots, k$. At each timestep $t$, the process involves: (1) *proposing* a sample $x_t^{(i)} \sim q(x_t \mid x_{<t}^{(i)})$, where $q$ is any locally normalized distribution; (2) *reweighting* by computing the importance weights $w_t^{(i)} = \phi(x_{1:t}^{(i)})/q(x_{1:t}^{(i)})$ where $\phi$ is an unnormalized potential satisfying $\phi(x_{1:n}) \propto p_{\mathrm{lm}}(x_{1:n} \mid \mathcal{C})$ at the final timestep $n$; and (3) *resampling* with replacement according to the normalized weights $w_t^{(i)}$ to obtain the new particles for the next iteration.

The performance of SMC depends heavily on the quality of the proposal distributions. LCD is a weak proposal for two reasons: it does not guarantee constraint satisfaction within a finite token budget, and it is a binary signal that does not have probabilistic information about future valid completions. As a result, SMC may require a large number of particles or expensive potentials to converge. In the following, we show that GCD (Section 3) addresses the first limitation, while P-GCD (Section 4) addresses the second.

## 3. Globally Constrained Decoding

To overcome the limitations of LCD, we introduce two constrained decoding distributions: (i) **globally constrained decoding (GCD)**, obtained by tensorizing constraints

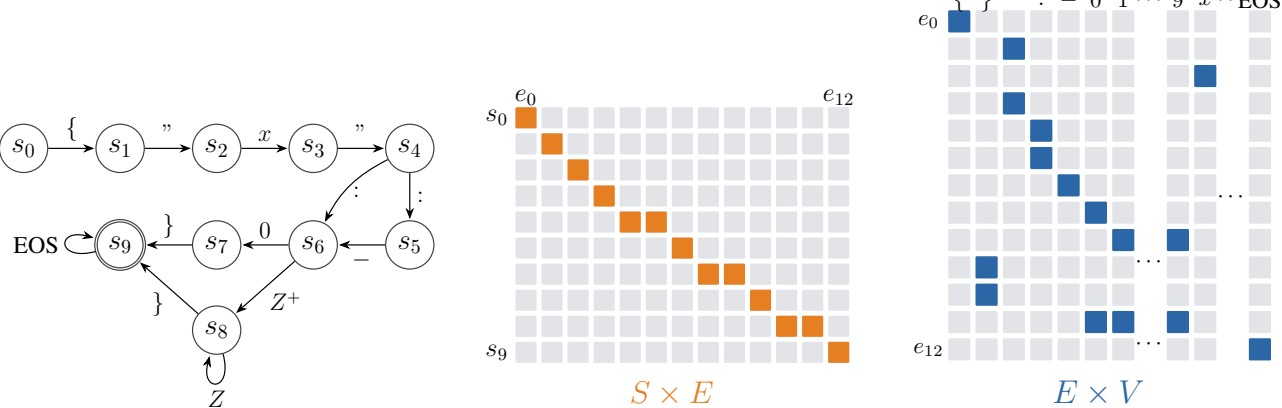

*Figure 1.* An illustration of an NFA corresponding to a JSON schema object of the form `{"x": <integer>}`, together with its tensorized representation. It consists of a transition matrix $T_{\text{out}}$ and an emission matrix $B_{\text{f}}$, which encode state transitions and symbol emissions, respectively. For simplicity, the incoming transition matrix $T_{\text{in}}$ is omitted in the figure.

specified as finite automata, and (ii) **probabilistic globally constrained decoding (P-GCD)**, obtained by circuit-multiplying the tensorized automaton with a hidden Markov model. While both can be used directly for constrained generation, we evaluate them as proposals (and potentials) within the SMC framework.

We first introduce **globally constrained decoding (GCD)**, a constrained generation distribution that masks tokens that cannot lead to a valid completion *within the token budget* $n$:

$$q_{\text{gcd}}(x_t \mid x_{<t}) \propto p_{\text{lm}}(x_t \mid x_{<t}) \cdot \mathbf{1}\{x_{1:t} \in \mathcal{X}_{t,n}\}. \quad (2)$$

Recall that $\mathcal{X}_{t,n}$ is the set of prefixes that can be extended to a length-$n$ sequence satisfying $\mathcal{C}$. GCD differs from LCD (Equation 1) only in the masking criterion: it masks with respect to $\mathcal{X}_{t,n}$ (prefixes that admit a valid completion within the remaining token budget) rather than $\mathcal{X}_{t,*}$ (prefixes that admit a valid completion at any length). As a result, GCD guarantees that every generated length-$n$ sequence satisfies $\mathcal{C}$. We now show how to represent the valid set $\mathcal{X}_n$ of length-$n$ sequences satisfying $\mathcal{C}$ compactly and evaluate the indicator $\mathbf{1}\{x_{1:t} \in \mathcal{X}_{t,n}\}$ efficiently for all vocabulary tokens at each decoding step using tensor operations.

### 3.1. Tensorizing Finite Automata

We consider constraints expressible as *finite automata*, specifically nondeterministic finite automata (NFAs) and deterministic finite automata (DFAs) (Hopcroft et al., 2000). Such constraints arise in a wide range of constrained generation tasks, including JSON-schema-constrained generation (Geng et al., 2025) and agentic applications involving function calling (Yao et al., 2024).

**Definition 3.1** (Finite Automata (Hopcroft et al., 2000)). A *finite automaton* (FA) is a tuple $\mathcal{M} = (\mathcal{S}, \mathcal{V}, \delta, s_0, \mathcal{F})$, where $\mathcal{S}$ is a finite set of *states*, $\mathcal{V}$ is a finite set of *symbols*

(tokens of the LLM vocabulary), $\delta$ is a *transition function*, $s_0 \in \mathcal{S}$ is the *initial state*, and $\mathcal{F} \subseteq \mathcal{S}$ is the set of *accepting states*. $\mathcal{M}$ is a *nondeterministic finite automaton* (NFA) when $\delta : \mathcal{S} \times \mathcal{V} \to 2^{\mathcal{S}}$ maps each state-symbol pair to a set of possible next states, and a *deterministic finite automaton* (DFA) when $\delta : \mathcal{S} \times \mathcal{V} \to \mathcal{S}$ maps each state-symbol pair to a unique next state. A token sequence $x_{1:n}$ is accepted by $\mathcal{M}$ if there exists a state sequence $s_0, s_1, \ldots, s_n$ such that $s_i \in \delta(s_{i-1}, x_i)$ for all $1 \le i \le n$, and $s_n \in \mathcal{F}$.

Figure 1 illustrates an NFA accepting JSON strings of the form `{"x":<integer>}`. We tensorize a finite automaton by viewing it as a labeled directed graph and encoding it as a collection of matrices. Let the states be $\mathcal{S} = \{s_0, \ldots, s_{S-1}\}$, where $s_0$ is the initial state and $S = |\mathcal{S}|$ is the number of states, and let the alphabet be $\mathcal{V} = \{v_0, \ldots, v_{V-1}\}$, where $V = |\mathcal{V}|$ is the vocabulary size. We group transitions so that for each state pair $(s_i, s_j)$ there is at most one edge labeled by a subset of $\mathcal{V}$. Let $\mathcal{E} = \{e_0, \ldots, e_{E-1}\}$ denote the resulting edge set, where $E = |\mathcal{E}|$ and each edge $e_k$ is associated with a source state $s_i$, a destination state $s_j$, and a subset of symbols that trigger the transition.

**Definition 3.2** (Tensorization of Finite Automata). We encode a finite automaton using three binary matrices:

$$T_{\text{out}} \in \{0,1\}^{S \times E}, \ T_{\text{in}} \in \{0,1\}^{E \times S}, \ B_{\text{f}} \in \{0,1\}^{E \times V},$$

where $T_{\text{out}}$ and $T_{\text{in}}$ are the source and destination incidence matrices, and $B_{\text{f}}$ encodes edge labels, with the subscript f representing FA. Specifically, for each edge $e_k$ from $s_i$ to $s_j$, set $T_{\text{out}}[i, k] = 1$ and $T_{\text{in}}[k, j] = 1$; and for each symbol $v_\ell$ that labels $e_k$, set $B_{\text{f}}[k, \ell] = 1$. The initial state $s_0$ and the accepting set $\mathcal{F}$ are represented by indicator vectors $\pi_{\text{f}}, f \in \{0,1\}^S$, where $\pi_{\text{f}} = (1, 0, \ldots, 0)^\top$ and $f_i = \mathbf{1}\{s_i \in \mathcal{F}\}$.

Figure 1 illustrates the resulting tensorized representation of the example NFA.

### 3.2. Inference with Tensorized Automata

We now show how to compute $q_{\text{gcd}}$ efficiently using this tensorized representation. To evaluate the indicator $\mathbf{1}\{x_{1:t} \in \mathcal{X}_{t,n}\}$, we must determine whether the partial sequence $x_{1:t}$ can be completed to a valid length-$n$ sequence. This requires answering two questions: (i) for each candidate next token $x_t \in \mathcal{V}$, which automaton states are reachable after consuming $x_{1:t-1}$; and (ii) from each such state, whether there exists a continuation of length $n - t$ that reaches an accepting state.

**Forward Process.** The forward process computes the set of FA states reachable after observing the prefix $x_{1:t-1}$, jointly for all candidate next tokens $x_t \in \mathcal{V}$. Intuitively, this corresponds to propagating state reachability along the FA transitions. We maintain a forward message $\alpha_t \in \{0,1\}^{S \times V}$, where $\alpha_t(s,v) = 1$ indicates that, after processing $x_{1:t-1}$, the FA can transition to state $s$ if the next token is $v$. Once a token $x_t$ is generated, the realized state is the column $\alpha_t(\cdot, x_t) \in \{0,1\}^S$. Thus the forward message can be computed recursively during autoregressive generation. Given the realized state $\alpha_{t-1}(\cdot, x_{t-1})$, the forward message at time $t$ is computed for all $v \in \mathcal{V}$ in parallel via:

$$\alpha_t(\cdot, \cdot) = \mathbf{1}\left\{T_{\text{in}}^\top \left(T_{\text{out}}^\top \alpha_{t-1}(\cdot, x_{t-1}) \circ B_{\text{f}}\right)\right\}, \quad (3)$$

where $T_{\text{out}}, T_{\text{in}}, B_{\text{f}}$ are defined in Definition 3.2, $\circ$ denotes element-wise multiplication with broadcasting along the vocabulary dimension, and $\mathbf{1}\{\cdot\}$ denotes the element-wise indicator. The base case is $\alpha_0(\cdot) = \pi_{\text{f}}$, corresponding to the initial automaton state at timestep $t = 0$.

**Backward Process.** During generation, we require that each prefix $x_{1:t}$ can be extended to a valid length-$n$ sequence satisfying the constraint. To enforce this, we maintain backward messages $\beta_t \in \{0,1\}^S$ where $\beta_t(s) = 1$ indicates that $s$ can reach an accepting state in exactly $n - t$ steps. The recursion is initialized at the final step with $\beta_n = f$, the indicator vector of accepting states. Given $\beta_t$, the backward message at the previous timestep is computed by propagating reachability backward along the FA edges:

$$\beta_{t-1} = \mathbf{1}\left\{T_{\text{out}}\left((T_{\text{in}}\beta_t) \circ (B_{\text{f}}\mathbf{1}_V)\right)\right\}, \quad (4)$$

where $\mathbf{1}_V \in \mathbb{R}^V$ is the all-ones vector, so $B_{\text{f}}\mathbf{1}_V$ sums each row of $B_{\text{f}}$ over the vocabulary dimension.

**Constraint Mask.** Combining forward and backward information yields a mask over the vocabulary: a token $v$ is feasible at step $t$ if and only if it reaches a state that admits a valid continuation.

$$m_t = \mathbf{1}\left\{\alpha_t^\top \beta_t\right\} \in \{0,1\}^V.$$

Tokens with $m_t(v) = 0$ are masked out at step $t$, guaranteeing that the generated sequence satisfies $\mathcal{C}$ within $n$ tokens. This yields an efficient algorithm for evaluating $\mathbf{1}\{x_{1:t} \in \mathcal{X}_{t,n}\}$ and hence computing $q_{\text{gcd}}(x_t \mid x_{<t})$.

## 4. Probabilistic Globally Constrained Decoding

While GCD guarantees constraint satisfaction within $n$ tokens, its mask is purely binary: a token is permitted if and only if it admits some valid completion, regardless of how likely that completion is under $p_{\text{lm}}$. This is the same limitation we identified for LCD in Section 2. To address this limitation, inspired by prior work (Zhang et al., 2023), we augment GCD with probabilistic information by distilling a hidden Markov model (HMM) $p_{\text{hmm}}$ to approximate $p_{\text{lm}}$ and combining it with the FA, yielding **probabilistic globally constrained decoding (P-GCD)**.

### 4.1. FA as HMM

We first show that FA naturally induces an HMM. Below we recap the definition of an HMM using tensor operations. The classic definition of an HMM is in Appendix B.

**Definition 4.1** (Hidden Markov Model (Rabiner & Juang, 1986))**.** A *hidden Markov model* (HMM) defines a joint distribution over an observed sequence $x_{1:n}$ and latent states $z_{1:n}$, where $z_t \in \{1, \ldots, H\}$ and $x_t \in \mathcal{V}$. It is parameterized by an initial distribution $\pi_{\text{h}} \in \mathbb{R}^H$, a transition matrix $A_{\text{h}} \in \mathbb{R}^{H \times H}$, and an emission matrix $B_{\text{h}} \in \mathbb{R}^{H \times V}$. The forward message $\alpha_t \in \mathbb{R}^H$, defined by $\alpha_t(i) = p_{\text{hmm}}(x_{1:t}, z_t = i)$, satisfies the recursion

$$\alpha_t = (A_{\text{h}}^\top \alpha_{t-1}) \circ B_{\text{h}}(\cdot, x_t), \quad \alpha_1 = \pi_{\text{h}} \circ B_{\text{h}}(\cdot, x_1),$$

where $\circ$ is element-wise multiplication, $B_{\text{h}}(\cdot, x_t)$ denotes the $x_t$-th column of $B_{\text{h}}$, and the subscript h represents HMM. The marginal likelihood of the prefix is then $p_{\text{hmm}}(x_{1:t}) = \sum_i \alpha_t(i)$.

Comparing the HMM forward recursion in Definition 4.1 with the FA forward recursion in Equation 3, we observe that they share the same algebraic structure. In fact, the tensorized FA implicitly defines an HMM whose hidden states are the edges of the FA.

**Definition 4.2** (Edge HMM induced by an FA)**.** The tensorized FA $(T_{\text{out}}, T_{\text{in}}, B_{\text{f}}, \pi_{\text{f}}, f)$ from Definition 3.2 induces an HMM whose hidden states correspond to FA edges. The transition and emission matrices are

$$A_{\text{f}} = T_{\text{in}}T_{\text{out}} \in \{0,1\}^{E \times E}, \quad B_{\text{f}} \in \{0,1\}^{E \times V},$$

where $A_{\text{f}}[k, k'] = 1$ if and only if the destination state of edge $e_k$ is the source state of edge $e_{k'}$. Overloading notation

from Definition 3.2, we redefine the initial-state and accept-state vectors in edge space as

$$\pi_{\mathrm{f}} := T_{\mathrm{out}}^{\top} \pi_{\mathrm{f}} \in \{0,1\}^E, \quad f := T_{\mathrm{in}} f \in \{0,1\}^E,$$

where the right-hand $\pi_{\mathrm{f}}$ and $f$ are the state-indexed versions from Definition 3.2; henceforth $\pi_{\mathrm{f}}$ and $f$ refer to their edge-lifted versions.

Since $A_{\mathrm{f}}$ and $B_{\mathrm{f}}$ contain binary values rather than probabilities, this induced HMM has unnormalized parameters. Rather than normalizing the transition and emission matrices, we compute unnormalized prefix probabilities and the corresponding normalization constants via forward and backward processes similar to Equations 3 and 4, enabling efficient likelihood evaluation and sampling from $p_{\mathrm{fa}}$.

**Theorem 4.3** (Distribution induced by an FA). *The induced HMM from Definition 4.2 yields a distribution*

$$p_{\mathrm{fa}}(x_{1:n}) \propto \left| \left\{ z_{1:n} : \begin{array}{l} z_{1:n} \text{ is an accepting path} \\ \text{producing } x_{1:n} \end{array} \right\} \right|.$$

*That is, each accepted sequence is weighted by its number of accepting paths, while unaccepted sequences have probability 0. For a DFA, every accepted sequence has exactly one accepting path, so $p_{\mathrm{fa}}$ is uniform over $\mathcal{X}_n$.*

### 4.2. The Product Distribution

The classical HMM (Definition 4.1) and the FA-induced HMM (Definition 4.2) share the same *circuit structure*, so applying circuit multiplication algorithms (Shen et al., 2016; Vergari et al., 2021; Loconte et al., 2025), their product

$$p_{\mathrm{prod}}(x_{1:n}) \propto p_{\mathrm{hmm}}(x_{1:n}) \cdot p_{\mathrm{fa}}(x_{1:n})$$

is a single HMM. Theorem 4.4 gives the explicit construction. As a corollary, when the FA is deterministic, $p_{\mathrm{fa}}$ is uniform over $\mathcal{X}_n$ (Theorem 4.3), and the product reduces to the constrained conditional

$$p_{\mathrm{prod}}(x_{1:n}) = p_{\mathrm{hmm}}(x_{1:n} \mid \mathcal{C}),$$

where $\mathcal{C}$ is the constraint represented by the DFA.

**Theorem 4.4** (Product HMM). *Let $(\pi_{\mathrm{h}}, A_{\mathrm{h}}, B_{\mathrm{h}})$ be the parameters of the language-model HMM and $(\pi_{\mathrm{f}}, A_{\mathrm{f}}, B_{\mathrm{f}})$ those of the FA-induced HMM. Their product distribution $p_{\mathrm{prod}} \propto p_{\mathrm{hmm}} \cdot p_{\mathrm{fa}}$ admits a representation as an HMM whose hidden state space is the Cartesian product of the two factors, with parameters*

$$\pi = \pi_{\mathrm{h}} \otimes \pi_{\mathrm{f}}, \quad A = A_{\mathrm{h}} \otimes A_{\mathrm{f}}, \quad B = B_{\mathrm{h}} \otimes_{\mathrm{row}} B_{\mathrm{f}},$$

*where $\otimes$ is the Kronecker product and $\otimes_{\mathrm{row}}$ denotes the row-wise Kronecker product, applied independently for each token: for each $v \in \mathcal{V}$, $B[:, v] = B_{\mathrm{h}}[:, v] \otimes B_{\mathrm{f}}[:, v]$.*

---

**Algorithm 1** Next-token probability for constrained generation

**Input:** prefix $x_{1:t-1}$, maximum sequence length $n$
  HMM $p_{\mathrm{hmm}}(A_{\mathrm{h}}, B_{\mathrm{h}}, \pi_{\mathrm{h}})$ with hidden size $H$
  FA $p_{\mathrm{fa}}(A_{\mathrm{f}}, B_{\mathrm{f}}, \pi_{\mathrm{f}}, f)$ with hidden size $E$
**Output:** $p_{\mathrm{prod}}(x_t{=}v, x_{<t})$ for any token $v$ in the vocabulary
*Convention:* $\alpha_t(z) = p_{\mathrm{prod}}(x_{<t}, z_t{=}z)$ (before-emission), reshaped in $\mathbb{R}^{H \times E}$; emission column $B(:, x) = B_{\mathrm{h}}(:, x) B_{\mathrm{f}}(:, x)^{\top}$.

**Step 1: Precompute backward messages**
**if** $t = 1$ **then**
  $\beta_n = \mathbf{1}_H f^{\top}$       ▷ accept indicator; exactly $n{-}t$ steps
  $input = B_{\mathrm{h}} B_{\mathrm{f}}^{\top}$       ▷ emission marginalized over vocab
  **for** $k$ **from** $n$ **to** 2 **do**
    $\beta_{k-1} = \left( A_{\mathrm{f}} \left( A_{\mathrm{h}}(input \circ \beta_k) \right)^{\top} \right)^{\top}$
  **end for**
  $\gamma = \pi_{\mathrm{f}}^{\top} \left( \pi_{\mathrm{h}}^{\top} \beta_1 \right)^{\top}$              ▷ total probability mass
**end if**

**Step 2: Compute prefix marginal**
**if** $t = 1$ **then**
  $\alpha_1 = \dfrac{1}{\gamma} (\pi_{\mathrm{h}} \otimes \pi_{\mathrm{f}})$
**else**
  $\alpha_t = \left( A_{\mathrm{f}}^{\top} \left( A_{\mathrm{h}}^{\top}(\alpha_{t-1} \circ B(:, x_{t-1})) \right)^{\top} \right)^{\top}$ ▷ emit $x_{t-1}$, *then*
  transition
**end if**
**return** $\mathrm{diag}\left( B_{\mathrm{f}}^{\top} \left( B_{\mathrm{h}}^{\top} (\alpha_t \circ \beta_t) \right)^{\top} \right)$

---

**Tensor Operations.** Rather than explicitly materializing the Kronecker-product matrices that define $p_{\mathrm{prod}}$, we operate directly on the factored representation. Specifically, we reshape each message in $\mathbb{R}^{HE}$ as a matrix in $\mathbb{R}^{H \times E}$ and apply the transition via the Kronecker-vec identity (Zhang et al., 2025): for any $u \in \mathbb{R}^{H \times E}$,

$$(A_{\mathrm{h}} \otimes A_{\mathrm{f}})u = \left( A_{\mathrm{f}}(A_{\mathrm{h}}u)^{\top} \right)^{\top} \tag{5}$$

where the left-hand side identifies $u$ with its vectorization. This reduces the cost of one transition step from $O((HE)^2)$ to $O(H^2 E + HE^2)$. Analogous tensor operations handle the emission (where $B_{\mathrm{h}} \otimes_{\mathrm{row}} B_{\mathrm{f}}$ acts column-wise) and the initial distribution (where $\pi_{\mathrm{h}} \otimes \pi_{\mathrm{f}}$ is a Kronecker product of vectors and never needs to be materialized).

**Autoregressive Generation.** To sample from $p_{\mathrm{prod}}$, we need next-token probabilities $p_{\mathrm{prod}}(x_t \mid x_{<t})$ at each step. Although $p_{\mathrm{prod}}$ has an HMM representation (Theorem 4.4), that HMM has unnormalized parameters, so the standard forward algorithm alone does not yield conditional next-token probabilities $p_{\mathrm{prod}}(x_t \mid x_{<t})$. We instead normalize on the fly: we precompute backward messages $\beta_t \in \mathbb{R}^{HE}$ to store the total probability mass of length-$(n - t)$ continuations

under $p_{\mathrm{prod}}$ given hidden state at time $t$, and combine them with forward marginals to obtain next-token probabilities at each step. Algorithm 1 gives the pseudocode.

The $\beta_t$ here is the real-valued analog of the binary backward message in Section 3: there, $\beta_t(s)$ indicated whether state $s$ could reach an accept state in $n - t$ steps; here, $\beta_t(z)$ measures the total probability mass of continuations reaching accept. Per step, computing $\beta_t$ costs $O(H^2E + HE^2 + HEV)$, which is quadratic in each of $H$ and $E$, linear in the vocabulary size $V$. Since forward and backward messages can be cached across timesteps, autoregressive decoding remains efficient.

### 4.3. P-GCD: Probabilistic Proposal and Potential

By leveraging the HMM's tractable approximation of the constrained likelihood, we introduce P-GCD, which provides both an SMC proposal and an SMC potential. As a proposal, P-GCD reweights the LM distribution at each step by the HMM's constrained next-token probability:

$$q_{\mathrm{pgcd}}(x_t \mid x_{<t}) \propto p_{\mathrm{lm}}(x_t \mid x_{<t})^w \cdot p_{\mathrm{hmm}}(x_t \mid x_{<t}, \mathcal{C})^{1-w},$$

where $w$ is a hyper-parameter to be tuned, and $p_{\mathrm{hmm}}(x_t \mid x_{<t}, \mathcal{C})$ is computed via $p_{\mathrm{prod}}(x_t \mid x_{<t})$ as in Algorithm 1.

P-GCD also serves as a potential, weighting particles by the LM's prefix probability times the HMM's look-ahead estimate that the prefix admits a constraint-satisfying completion:

$$\phi_{\mathrm{pgcd}}(x_{1:t}) \propto p_{\mathrm{lm}}(x_{1:t}) \cdot p_{\mathrm{hmm}}(\mathcal{C} \mid x_{1:t})$$
$$\propto p_{\mathrm{lm}}(x_{1:t}) \cdot p_{\mathrm{hmm}}(x_{1:t} \mid \mathcal{C}) / p_{\mathrm{hmm}}(x_{1:t}),$$

where the second form follows from Bayes' rule (the constant $p_{\mathrm{hmm}}(\mathcal{C})$ is absorbed into $\propto$) and is operationally convenient: $p_{\mathrm{hmm}}(x_{1:t} \mid \mathcal{C})$ is the prefix marginal under $p_{\mathrm{prod}}$ (Algorithm 1), and $p_{\mathrm{hmm}}(x_{1:t})$ is the prefix marginal under the unconstrained HMM. In Section 5, we show that both the proposal and the potential improve SMC convergence, reducing the number of particles required.

## 5. Experiments

### 5.1. Baselines

We compare the GCD proposal $q_{\mathrm{gcd}}$, the P-GCD proposal $q_{\mathrm{pgcd}}$, and the P-GCD potential $\phi_{\mathrm{pgcd}}$ within the SMC framework against the following baselines:

- **Rejection sampling (RS).** Repeatedly sample from $p_{\mathrm{lm}}(x_{1:n})$ until a sequence satisfying $\mathcal{C}$ is obtained, yielding exact samples from $p_{\mathrm{lm}}(x_{1:n} \mid \mathcal{C})$. This serves as the ground-truth distribution.
- **Locally constrained decoding (LCD).** At each step, masks tokens that immediately violate the constraint,

renormalizes the remaining probability mass, and samples from the resulting distribution.
- **LCD + SMC (Loula et al., 2025).** Uses LCD as the proposal distribution within an SMC framework to approximate the target conditional distribution $p_{\mathrm{lm}}(x_{1:n} \mid \mathcal{C})$.

### 5.2. Function Calling

**Task and Dataset.** We first evaluate on the xlam-function-calling-60k (xLAM) dataset (Liu et al., 2024). Each example from the dataset provides a set of "functions" that the language model can call to retrieve information needed to answer a user query. We first filter out examples where the ground-truth answer does not satisfy the given function specification (e.g., argument type mismatch). From the remaining examples, we randomly sample 1,000 instances as a test set. We evaluate two function-calling formats: **JSON** and **Python-like** syntax. For example, given the user query "What's the weather like in NYC today?", the model should generate the function call `{"name": "get_weather", "location": "NYC"}` in JSON format or `get_weather(location="NYC")` in Python-like syntax.

**Metric.** The model may only call functions provided in the example, and an output is considered correct if it (i) follows the required syntax (i.e., is parsable) and (ii) calls the correct function with the correct arguments, matching the ground-truth answer.

**Constraints and FA Construction.** Each function definition is provided as a JSON schema, which we convert into an equivalent FA that accepts exactly the set of valid JSON strings conforming to the schema. We use standard JSON Schema specifications [2] to construct the FA, first building primitive types such as literals and strings, then composite types including arrays and objects, and finally composing them into function-calling schemas. We use the AUTOMATA library for FA construction [3] and operations such as union, option, and concatenation (Evans & Robson, 2023). We construct FAs for Python-like function-calling syntax in a similar manner.

**Experiments and Results.** We use Llama-3.1-8B-Instruct as the base model and follow the pipeline of Zhang et al. (2024) to distill Monarch HMMs (Zhang et al., 2025) with $2^{14}$ hidden states (i.e., $2^{21}$ FLOPs per token during inference). As shown in Figure 2, for both JSON and Python-like formats, (P-)GCD achieves substantially higher accuracy than LCD at the same number of particles, converging to the ground-truth distribution (rejection sampling) much faster. In particular, P-GCD exhibits a faster convergence rate than

---

[2] https://json-schema.org/
[3] https://github.com/caleb531/automata

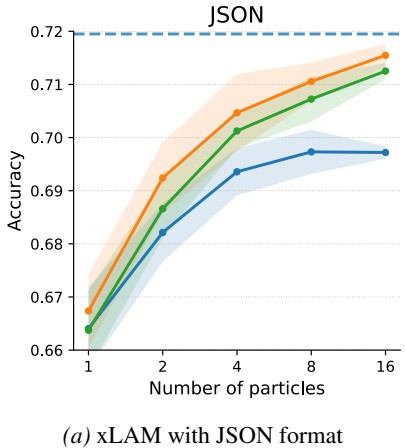

*(a)* xLAM with JSON format

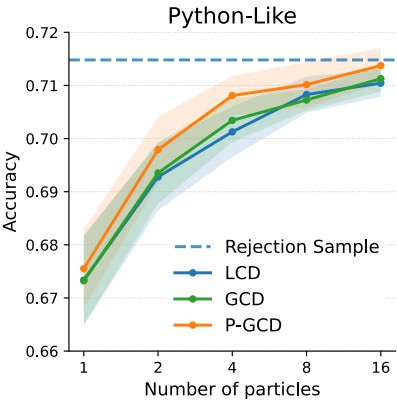

*(b)* xLAM with Python-like format

*Figure 2.* Accuracy (%) on xLAM as a function of the number of SMC particles comparing proposals LCD, GCD, and P-GCD.

GCD, illustrating the benefit of the additional probabilistic information provided by the HMM. We additionally report constraint satisfaction rates at fixed particle budgets in Table 1. LCD can fail to satisfy the constraint even at moderate particle budgets, partially explaining its lower accuracy compared to GCD and P-GCD.

*Table 1.* Constraint satisfaction rates (%).

| Method | JSON | | Python-like | |
|---|---|---|---|---|
| | $k{=}4$ | $k{=}16$ | $k{=}4$ | $k{=}16$ |
| LCD | 95.26 | 95.40 | 98.12 | 99.20 |
| GCD | 100.0 | 100.0 | 100.0 | 100.0 |
| P-GCD | 100.0 | 100.0 | 100.0 | 100.0 |

**Runtime Comparison.** We measure wall-clock latency on xLAM. Table 2 shows that LCD and GCD incur only modest overhead relative to unconstrained generation ($1.08\times$ and $1.10\times$, respectively). P-GCD is more expensive due to the additional HMM computations, ranging from $3.11\times$ to $10.2\times$ slowdown depending on the HMM size. HMM distillation is a one-time offline procedure consisting of (i) sampling data from the base model, which requires approximately 2 GPU hours on H100 to generate 2M samples using vLLM, and (ii) training the HMM, which takes approximately 0.2 to 3 GPU hours for hidden sizes ranging from 1024 to 16384.

**Sensitivity to HMM size.** Table 3 reports xLAM accuracy for different HMM hidden sizes. We find that even relatively small HMMs provide useful guidance yet larger HMMs consistently yield better performance; the advantage is mostly exhibited in low-particle regimes, while the performance gap diminishes as the number of particles increases. This suggests that P-GCD is robust to the quality of the HMM

*Table 2.* Wall-clock latency and slowdown compared to unconstrained generation on xLAM using Llama-3.1-8B-Instruct, reported as seconds per token (mean $\pm$ standard deviation over 3 seeds). Measured on a single NVIDIA H100 GPU over 100 examples with batch size 16; $H$ denotes the HMM's hidden size.

| Method | ms/token | Slowdown |
|---|---|---|
| Unconstrained | $1.69 \pm 0.01$ | $1.00\times$ |
| LCD | $1.83 \pm 0.02$ | $1.08\times$ |
| GCD | $1.86 \pm 0.04$ | $1.10\times$ |
| P-GCD ($H = 1024$) | $5.26 \pm 0.01$ | $3.11\times$ |
| P-GCD ($H = 4096$) | $8.23 \pm 0.63$ | $4.87\times$ |
| P-GCD ($H = 16384$) | $17.30 \pm 0.39$ | $10.2\times$ |

*Table 3.* Accuracy (%) on xLAM with JSON format comparing LCD and P-GCD with different HMM's hidden size.

| Method | $k{=}1$ | $k{=}2$ | $k{=}4$ | $k{=}8$ | $k{=}16$ |
|---|---|---|---|---|---|
| LCD | 66.4 | 68.2 | 69.3 | 69.7 | 69.7 |
| P-GCD ($H{=}1024$) | 65.4 | 68.6 | 70.2 | 71.1 | 71.5 |
| P-GCD ($H{=}16384$) | 66.5 | 69.2 | 70.5 | 71.5 | 71.6 |

approximation and does not require highly accurate models to be effective.

### 5.3. Keyword-based Generation

**Task and Dataset.** We also evaluate (P-)GCD on the CommonGen benchmark (Lin et al., 2020). Each test example provides 3 to 5 concepts (keywords) as input, and the goal is to generate a natural sentence that incorporates all concepts, allowing for any inflections. For example, given *"car"*, *"snow"*, and *"drive"*, both *"a man drives a car on a snow-covered road"* and *"the car drove through the snow"* are considered acceptable.

**Metric.** We use BLEU (Papineni et al., 2002) as the evaluation metric. Since generation involves random sampling

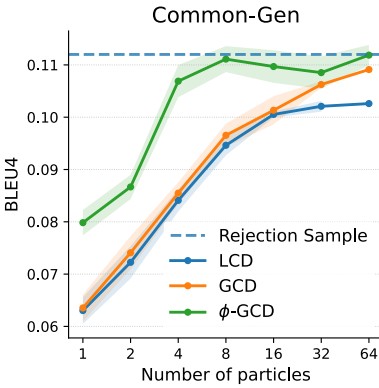

*Figure 3.* BLEU-4 on CommonGen as a function of the number of SMC particles. LCD and GCD configurations use the LM likelihood $p_{lm}$ as the potential, with proposals $q_{lcd}$ and $q_{gcd}$ respectively; $\phi$-GCD uses $q_{pgcd}$ as the proposal and $\phi_{pgcd}$ as the potential.

from the proposal distribution with temperature 1.0, we report the expected BLEU score, weighted over particles produced by SMC sampling.

**Experiments and Results.** In this setting, we evaluate P-GCD as a potential function by incorporating $\phi_{pgcd}$ into SMC sampling with $q_{pgcd}$ as the proposal. We use GPT2-Large as the base language model and adopt the HMM checkpoint released by Zhang et al. (2023). As shown in Figure 3, using $\phi_{pgcd}$ in addition to the $q_{pgcd}$ proposal further improves convergence, yielding a substantial advantage over the LCD proposal.

### 5.4. Text-to-SQL Generation

**Task and Dataset.** We further evaluate on Spider (Yu et al., 2018), a text-to-SQL benchmark featuring SQL syntax constraints. Following the experimental setup of xLAM, we randomly sample 500 examples for evaluation and use the remaining data to distill the HMM for P-GCD.

**Metric.** We report execution accuracy: a generated SQL query is correct if executing it on the target database returns the same result as the reference query.

**Results.** Table 4 reports results for different particle budgets. Both GCD and P-GCD improve upon LCD across a range of particle budgets, with the largest improvements observed at moderate values of $k$. These results suggest that the benefits of GCD extend beyond function calling to more complex structured generation tasks.

### 5.5. NFA vs. DFA

**Throughout this paper, we treat finite automata (FAs) generically, without distinguishing between nondeterministic FAs**

*Table 4.* Accuracy (%) on Spider as a function of the number of SMC particles using different proposal distributions.

| Method | $k = 1$ | $k = 4$ | $k = 8$ |
|--------|---------|---------|---------|
| LCD    | 59.8    | 65.4    | 67.2    |
| GCD    | 59.0    | 66.8    | 69.0    |
| P-GCD  | 56.2    | 67.9    | 69.1    |

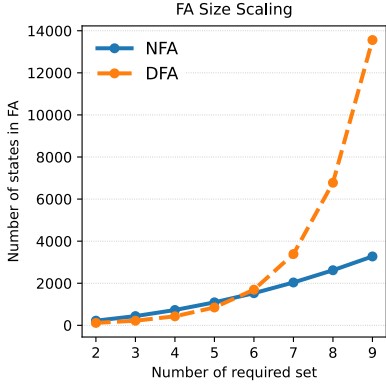

*Figure 4.* NFA and DFA state counts as a function of required set, illustrating the exponential succinctness gap.

(NFAs) and deterministic FAs (DFAs). In our experiments, xLAM and CommonGen use DFAs, while text-to-SQL uses NFAs. Here we present a more specific case study illustrating that NFAs can be exponentially more succinct than DFAs. Continuing with the function-calling task, we consider the following definition of function add_vec_n and impose a JSON constraint on the function-call output:

```
{
  "name": "add_vec_n",
  "parameters": {
    "type": "object",
    "properties": {
      "x1": {"type": "integer"},
      ...
      "xn": {"type": "integer"},
      "y1": {"type": "integer"},
      ...
      "yn": {"type": "integer"},
    },
    "anyOf": [
      { "required": ["x1", "y1"] },
      ...
      { "required": ["xn", "yn"] },
    ]
  }
}
```

Figure 4 reports the sizes of the NFAs and the theoretically minimal DFAs encoding this JSON schema as a function of $n$. As $n$ increases, the NFA size grows linearly, whereas the DFA size grows exponentially, rendering DFAs impractical in this setting. In real-world agentic applications, function definitions often exhibit complex dependencies among parameters, where such DFA blow-ups may arise in practice.

*Table 5.* Number of states in the SQL constraint FA used for Spider. OOM denotes out-of-memory during DFA construction.

| Granularity | NFA states | DFA states |
|---|---|---|
| Character-level | 7,999 | 57,263 |
| Token-level | 31,858 | CPU OOM |

This example highlights the importance of NFA support for constrained decoding.

Furthermore, the compactness advantage of NFAs also manifests in practical structured generation tasks. For the SQL grammar used in our Spider experiments, the corresponding NFAs are substantially smaller than their DFA counterparts (Table 5). At the character level, the NFA has about 8k states compared to about 57k for the equivalent DFA; at the token level, the NFA has about 32k states, while DFA construction exceeded available CPU memory. These results motivate operating directly on NFAs.

## 6. Related Work

**Locally Constrained Decoding.** There is an extensive line of work on locally constrained decoding (Shin et al., 2021; Scholak et al., 2021; Poesia et al., 2022; Ugare et al., 2025; Park et al., 2025), with a primary focus on enforcing hard constraints efficiently during autoregressive generation. These approaches typically operate by masking out invalid next tokens based on the current prefix, thereby ensuring local constraint satisfaction at each decoding step. Existing frameworks differ mainly in how constraints are represented: for example, Outlines (Willard & Louf, 2023) uses regular expressions and their corresponding DFA representations, while more recent systems such as XGrammar (Dong et al., 2025) rely on context-free grammars. Notably, most of these systems are optimized for CPUs executions. In contrast, we propose the first tensorized representation of automaton-based constraints together with GPU-native inference algorithms. More importantly, our approach guarantees the constraint will be satisfied upon termination.

**Approximate Inference for Constrained Generation.** Recent work has formulated constrained generation as approximate inference of the conditional distribution. Prior approaches include MCMC-based methods (Qin et al., 2022; Gonzalez et al., 2026), which iteratively refine samples toward the target distribution; amortized inference methods (Hu et al., 2024), which learn auxiliary approximations to the conditional distribution; and sequential Monte Carlo (SMC) methods (Loula et al., 2025; Lipkin et al., 2025), which maintain weighted particles and asymptotically recover the target distribution through importance weighting and resampling.

**Controllable Generation with Auxiliary Models.** Several works incorporate auxiliary models or signals to guide constrained or controllable generation. ABS (Collura et al., 2025) re-weights language model logits using the distance to an accepting state in a DFA, thereby encouraging progress toward constraint satisfaction. GeLaTo (Zhang et al., 2023) is the first to use a hidden Markov model to capture probabilistic information from the language model, but it is limited to keyword-based constraints. Its follow-up work, Ctrl-G (Zhang et al., 2024), combines DFA constraints with HMM guidance; however, its algorithm does not handle NFAs. As discussed in Section 5.5, DFAs can be exponentially larger than equivalent NFAs, making DFA-based approaches infeasible in many real-world settings. Methods such as GeDi (Krause et al., 2021), FUDGE (Yang & Klein, 2021), and NADO (Meng et al., 2022) train auxiliary neural classifiers to steer generation, but they provide no guarantees on exact constraint satisfaction.

## 7. Conclusion

This work aims to mitigate the sampling bias introduced by locally constrained decoding for constrained generation from large language models. In conclusion, we present **(probabilistic) globally constrained decoding**, an algorithmic approach for constructing proposal distributions and potential functions within the sequential Monte Carlo sampling framework. By tensorizing constraints represented as finite automata and (optionally) multiplying them with hidden Markov models, we can obtain tractable proposals and potentials that capture not only the logical structure of the constraints but also the probabilistic information from the language model. Empirical evaluations on xLAM, CommonGen and, Spider demonstrate that, compared to locally constrained decoding–based proposals, our approach not only guarantees constrained satisfaction within any given token budget but also achieves substantially better convergence in SMC sampling with far fewer particles.

## Acknowledgments

This work is supported in part by ARO (W911NF-21-1-0125), ONR (N00014-23-1-2159), and the CZ Biohub. This work is funded in part by the DARPA ANSR and CODORD programs under awards FA8750-23-2-0004 and HR00112590089, and gifts from Cisco Research, Qualcomm, and Amazon.

## Impact Statement

This paper presents work whose goal is to advance the field of machine learning. There are many potential societal consequences of our work, none of which we feel must be specifically highlighted here

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

## A. Locally Constrained Decoding

To see why LCD distors the LM distribution, consider a simple example where $p_{\text{lm}}$ defines a uniform distribution of 01 sequences of length 3 (i.e., $p_{\text{lm}}(x_t \mid x_{<t}) = 1/2$ for $x_t \in 0, 1$). Let $\mathcal{C}$ require exactly one 1 in the sequence, so the valid sequences are 001, 010, and 100, each of which has probability $1/3$. Under locally constrained decoding equation 1, $p_{\text{lm}}(x_1 \mid \mathcal{C}) = 1/2$ for $x_1 = 0$ and $x_1 = 1$. For $x_1 = 0$, both $x_2 = 0$ (to 001) and $x_2 = 1$ (to 010) are possible, so $p_{\text{lm}}(x_2 \mid x_1 = 0, \mathcal{C}) = 1/2$ each, with $x_3$ deterministic (1 or 0). For $x_1 = 1$, only $x_2 = 0$ (to 100) is possible, so $p_{\text{lm}}(x_2 \mid x_1 = 1, \mathcal{C}) = 1$ and $x_3 = 0$. The resulting probabilities are $p_{\text{lm}}(001 \mid \mathcal{C}) = p_{\text{lm}}(010 \mid \mathcal{C}) = 1/4$ and $p_{\text{lm}}(100 \mid \mathcal{C}) = 1/2$. The issue is that local renormalization lacks the ability of "lookahead": as the model generates the first digit $x_1$, it cannot count globally how many examples satisfy the constraint and start with this digit, therefore, it equally splits probability mass between $x_1 = 0$ and $x_1 = 1$. After $x_1 = 1$, however, there is only one branch, capturing the full remaining mass. It cannot account for the global number of valid completions per prefix.

## B. Hidden Markov Models

A Hidden Markov Model (HMM) (Rabiner & Juang, 1986) represents a joint probability distribution over $n$ observed variables $x_{1:n}$ and $n$ hidden variables $z_{1:n}$. Specifically, for language modeling, $x_t$ represents the token at position $t$ and $z_t$ is the corresponding hidden state; $z_t$ takes values in $\{1, 2, \ldots, H\}$, where $H$ is the *number of hidden states*. An HMM models the joint distribution:

$$p(x_{1:n}, z_{1:n}) = p(x_1 \mid z_1) \cdot p(z_1) \cdot \prod_{2 \leq t \leq n} p(x_t \mid z_t) \cdot p(z_t \mid z_{t-1}).$$

In particular, the parameters of an HMM are given by the initial probability $p(z_1)$, the emission matrix $p(x_t \mid z_t)$ and the transition matrix $p(z_{t+1} \mid z_t)$; the number of parameters of HMMs grows quadratically with respect to $H$.

In the main text, we use matrices and vectors to represent the transition probabilities $A$, emission probabilities $B$, and initial probability $\pi$. Specifically, let

$$A \in \mathbb{R}^{H \times H}, \quad A(i, j) = p(z_{t+1} = j \mid z_t = i);$$
$$B \in \mathbb{R}^{H \times V}, \quad B(i, v) = p(x_t = v \mid z_t = i);$$
$$\pi \in \mathbb{R}^H, \quad \pi(i) = p(z_1 = i).$$

To perform inference on HMMs efficiently, we leverage the *Markov property*: $p(x_{\geq t} \mid z_t, x_{<t}) = p(x_{\geq t} \mid z_t)$. For example, we can efficiently compute $p(x_{\leq t}) = \sum_{z_t} p(x_{\leq t}, z_t)$ by the following recurrence relation, referred to as the *forward algorithm* (Rabiner & Juang, 1986):

$$p(x_{\leq t}, z_t) = p(x_t \mid z_t) \cdot \sum_{1 \leq z_{t-1} \leq h} p(z_t \mid z_{t-1}) \cdot p(x_{\leq t-1}, z_{t-1}).$$

In the tensor representation, we define the forward message at time $t$ as

$$\alpha_t \in \mathbb{R}^H, \qquad \alpha_t(i) \stackrel{\text{def}}{=} p(x_{\leq t}, z_t = i).$$

Therefore the forward algorithm can then be written compactly as the recursion

$$\alpha_t = \left(A^\top \alpha_{t-1}\right) \circ B(:, x_t),$$

where $\circ$ denotes element-wise multiplication and $B(:, x_t)$ denotes the column of $B$ corresponding to the observed token $x_t$. The base case is given by

$$\alpha_1 = \pi \circ B(:, x_1).$$

Therefore, the marginal is

$$p(x_{\leq t}) = \mathbf{1}_H^\top \alpha_t,$$

where $\mathbf{1}_H \in \mathbb{R}^H$ denotes the all-ones vector.

