# OpenReview forum: "Mitigating Bias in Locally Constrained Decoding via Tractable Proposals"
_ICML.cc/2026/Conference — ICML 2026 regular_

### Official Review · Reviewer_cRNx · 2026-03-07

**Soundness:** 3
**Presentation:** 3
**Significance:** 2
**Originality:** 3
**Overall Recommendation:** 4
**Confidence:** 2

**Summary:**

This paper focuses on an important issue in constrained generation: traditional locally constrained decoding (LCD) satisfies constraints by gradually masking illegal next tokens, but this local greedy strategy introduces sampling bias and does not guarantee that the final output meets global constraints under a limited token budget. To address this, the authors propose globally constrained decoding (GCD), which tensorizes finite automata to efficiently determine on a GPU whether a certain prefix can still be completed into a valid sequence within the remaining length. On this basis, the authors further introduce probabilistic GCD (P-GCD), which combines the tensorized automaton with HMM to construct proposals/potentials that contain both logical constraints and probabilistic information for SMC sampling. Experiments on xLAM function calling and CommonGen show that compared with LCD-based proposals, (P-)GCD converges faster to the target distribution with fewer particles under the same SMC settings.

**Compliance With Llm Reviewing Policy:**

Affirmed.

**Ethics Expertise Needed:**

["Other Expertise"]

**Final Justification:**

Thank you for your response. Most of my issues have been resolved, and I will improve my score.

**Key Questions For Authors:**

1. How sensitive is the performance of P-GCD to the hidden size and distillation quality of HMM? Is there an obvious performance-computation trade-off curve?
2. In addition to xLAM and CommonGen, is the method equally effective in more complex structured generation scenarios (such as SQL or code syntax constraints)?
3. The paper emphasizes faster convergence with fewer particles, but when the additional computation of the proposal/potential itself is included, what is the actual wall-clock advantage of P-GCD compared to LCD+SMC?
4. The discussion on NFA and DFA is interesting. Have the authors tried to systematically compare the actual efficiency differences between NFA-based and DFA-based implementations on real benchmarks, rather than just the scale of the number of states?

**Limitations:**

yes

**Strengths And Weaknesses:**

### Strengths

* The paper addresses the sampling bias problem of LCD in constrained generation, with a clear problem definition and strong practical significance, especially directly related to applications such as JSON schema and function calling.
* The core idea of GCD is relatively clear: changing from "locally valid" to "globally completable within the remaining budget", which is stricter than LCD and more in line with real decoding settings.
* Tensorizing finite automata and performing GPU-native inference is a good technical contribution, with both methodological and system implementation value. Figure 1 on Page 3 and the forward/backward derivation on Page 4 well illustrate this point.
* Combining HMM with GCD to construct P-GCD proposals/potentials enables the method to utilize not only logical information but also the probabilistic structure of the target distribution, which is more convincing than simple masking.
* The experimental results generally support the paper's claims. On xLAM, Figure 2 shows that GCD/P-GCD is significantly superior to LCD with the same number of particles; Table 1 also shows that GCD/P-GCD can achieve 100% constraint satisfaction, while LCD cannot.
* On CommonGen, the addition of P-GCD potential further improves convergence, indicating that both the proposal and potential designs are effective.

### Weaknesses

* The paper relies heavily on the SMC framework overall, and it is more like designing a better proposal/potential for SMC rather than proposing a completely independent new constrained decoding framework; the boundary of contributions can be stated more clearly.
* The empirical evaluation is not yet comprehensive. The tasks are mainly focused on xLAM and CommonGen, which are representative but still far from broader structured generation/code generation/SQL generation scenarios.
* The quality of HMM approximation is crucial for P-GCD, but the paper's sensitivity analysis on this point is still limited. For example, the impact of hidden size and distillation quality on the final performance is not expanded.
* The computational cost analysis can be more complete. The paper emphasizes that "fewer particles are needed to converge", but P-GCD itself also introduces additional tractable model computation, and the actual wall-clock tradeoff can be more explicit.
* Certain theoretical expressions are relatively compact, making the understanding threshold high for readers outside this field. In particular, the relationship between tensorization, circuit multiplication, and HMM product requires a more intuitive explanation.
* The impact statement is too brief, and the potential social impacts are basically not fully discussed.

---

> ### Author Rebuttal · Authors · 2026-03-31
>
> We thank the reviewer for the thoughtful and constructive feedback.
>
> > **W1: The boundary of contribution**
>
> Thank you for raising this important point. We would like to clarify that:
>
> 1. compared to LCD, GCD guarantees that constraints will always be satisfied within a given token budget, which (1) addresses a key need in real-world serving of LLMs and (2) can be applied as a *standalone decoding algorithm without SMC*.
> 2. further, by incorporating the probabilistic information from an HMM, P-GCD shows advantages over the LCD baseline without SMC; for example, in Figure 2, when k = 1 (no SMC), P-GCD > LCD
>
>
> > **W2: theoretical expressions are relatively compact … impact statement is too brief …**
>
> Thank you for your suggestions regarding the presentation. In the final version, we will include a more detailed, step-by-step proof sketch for the theoretical results, as well as visualizations illustrating how tensorized automata and HMMs are combined.
> We will also expand the impact statement to provide a more thorough discussion of how this work may influence the serving of LLMs and its potential (both positive and negative) societal consequences.
>
> > **Q1. How sensitive is the performance of P-GCD to the hidden size and distillation quality of HMM?**
>
> Thank you for raising this important question. We present a preliminary ablation here studying how the HMM quality affects the performance of P-GCD.
>
>
> | Hidden size | Acc (k=1) | Acc (k=2) | Acc (k=4) | Acc (k=8) | Acc (k=16) |
> |---|---|---|---|---|---|
> | LCD | 66.4 ± 0.7 | 68.2 ± 0.5 | 69.3 ± 0.4 | 69.7 ± 0.4 | 69.7 ± 0.1 |
> | 1024 | 65.4 ± 0.9 | 68.6 ± 0.7 | 70.2 ± 0.2 | 71.1 ± 0.6 | 71.5 ± 0.1 |
> | 16384 | 66.5 ± 0.8 | 69.2 ± 0.5 | 70.5 ± 0.4 | 71.5 ± 0.1 | 71.6 ± 0.1 |
>
> We find that
> 1. P-GCD shows advantage over the LCD baseline even with small HMMs, showing its robustness against the quality of HMMs.
> 2. The performance of P-GCD improves as the HMM size increases.
> 3. While larger HMMs always show stronger performance, the performance gap diminishes as the number of particles increases, showing an interesting trade-off.
>
> We will include a more comprehensive ablation study in the revision.
>
> > **Q2. is the method equally effective in more complex structured generation scenarios?**
>
> We agree that broader evaluation would strengthen the paper and further evaluate on text-to-SQL generation using the Spider benchmark [1], which involves more complex compositional constraints. Following the xLAM setup, we use a random split of 500 prompts for testing and the remainder for distilling a small HMM (hidden size 1024), which is used by P-GCD.
> | Method | Acc (k=1) | Acc (k=4) | Acc (k=8) |
> |---|---:|---:|---:|
> | LCD | 59.8 | 65.4 | 67.2 |
> | GCD | 59.0 | 66.8 | 69.0 |
> | P-GCD | 56.2 | 67.9 | 69.1 |
>
> In the text-to-SQL setting, (P-)GCD also shows improvement across different numbers of particles k.
>
> > **Q3. what is the actual wall-clock advantage of P-GCD compared to LCD+SMC?**
>
> For xLAM, we report wall-clock time per token for unconstrained generation, LCD, GCD, and P-GCD using Llama 3 8B. Results are averaged over 100 examples (batch size 16); wall-clock is reported as mean ± std over 3 seeds. H denotes the HMM hidden size. All experiments run on a single NVIDIA H100 GPU.
>
>
> | Method | ms / token | Slowdown vs unconstrained |
> |---|---|---|
> | Unconstrained | 1.69 ± 0.01 | 1.00× |
> | LCD | 1.83 ± 0.02 | 1.08× |
> | GCD | 1.86 ± 0.04 | 1.10× |
> | P-GCD (H=1024) | 5.26 ± 0.01 | 3.11× |
> | P-GCD (H=4096) | 8.23 ± 0.63 | 4.87× |
> | P-GCD (H=16384) | 17.30 ± 0.39 | 10.2× |
>
> 1. GCD and LCD add minimal overhead (<10%) over unconstrained generation.
> 2. In particular, GCD only introduces 2% overhead in addition to LCD while achieving significantly better performance.
> 3. P-GCD incurs higher per-token cost than GCD, but for tasks where semantic/probabilistic information about the constraints are important (e.g., CommonGen), it yields faster convergence as an SMC proposal. As shown in Fig. 3, P-GCD reaches ground-truth performance with k=4, matching GCD/LCD at k=64, reducing the number of inference calls to the base model by 16 times, which eventually roughly saves 4 times of total compute.
>
> > **Q4. Have the authors tried to systematically compare the actual efficiency differences between NFA-based and DFA-based implementations on real benchmarks?**
>
> In practical structured generation tasks (e.g., SQL or JSON/function-calling), constraints often involve optional fields and compositional patterns. These lead to state-space explosion when converted to DFAs.
>
> For example, in our text-to-SQL experiments (Spider), for encoding the minimal SQL syntax, NFAs are significantly more compact than their DFA counterparts.
>
> | Representation | NFA size | DFA size |
> |---|---|---|
> | Character-level | 7999 | 57263 |
> | Token-level | 31858 | Compilation OOM (CPU) |
>
> [1] Yu, T., et al. (2018). Spider: A Large-Scale Human-Labeled Dataset for Complex and Cross-Domain Semantic Parsing and Text-to-SQL Task.

---

> > ### Author Rebuttal · Reviewer_cRNx · 2026-04-03
> >
> > Thank you for your response. Most of my issues have been resolved, and I will improve my score.

---

### Official Review · Reviewer_oDaW · 2026-03-07

**Soundness:** 3
**Presentation:** 3
**Significance:** 3
**Originality:** 4
**Overall Recommendation:** 4
**Confidence:** 3

**Summary:**

This paper mitigates the sampling bias of locally-constrained decoding (LCD) in large language models by designing tractable proposals for Sequential Monte Carlo (SMC) sampling(). The authors introduce Globally Constrained Decoding (GCD) by tensorizing finite automata to guarantee constraint satisfaction within a fixed token budget. Furthermore, they develop Probabilistic GCD (P-GCD) by circuit-multiplying these automata with a distilled language-model Hidden Markov Model (HMM) to incorporate probabilistic lookahead(). Experiments demonstrate that these methods significantly improve SMC convergence efficiency and generation quality over LCD baselines.

**Compliance With Llm Reviewing Policy:**

Affirmed.

**Final Justification:**

The rebuttal addressed my main concerns and clarified several important points. In particular, the authors provided helpful additional evidence and discussion on the practical tradeoffs of the method. However, my second concern is not fully resolved, as the rebuttal still does not sufficiently analyze whether the proposed approach remains effective and necessary for larger, frontier-scale models. Overall, I find the paper potentially weakly acceptable, so I keep my score unchanged.

**Key Questions For Authors:**

1. How do the proposed methods perform on more complex, longer-form structured generation tasks compared to the relatively short xLAM and CommonGen benchmarks?
2. Are the complex P-GCD proposals still necessary and beneficial when applied to larger, frontier-class models, which may already naturally align with the required syntax?
3. Could the authors provide a direct empirical comparison of practical inference throughput and memory footprint against established parsers like Outlines or XGrammar?

**Limitations:**

yes

**Strengths And Weaknesses:**

**Strengths**
- Innovative FA Tensorization: The paper introduces a novel method to tensorize Finite Automata (FA) for efficient, GPU-native constraint execution, advancing beyond traditional CPU-based parsers.
- Elegant Probabilistic Integration: By recognizing that tensorized FAs share a circuit structure with Hidden Markov Models (HMMs), the authors creatively use circuit multiplication to construct the Probabilistic GCD (P-GCD) proposal, effectively capturing both strict logical rules and probabilistic lookahead.
- Guaranteed Global Constraints: Unlike greedy locally constrained decoding (LCD), the proposed Globally Constrained Decoding (GCD) mathematically guarantees constraint satisfaction within a fixed, finite token budget.
- Strong Empirical Results: On function-calling (xLAM) and commonsense generation (CommonGen) benchmarks, both GCD and P-GCD achieve faster Sequential Monte Carlo (SMC) convergence with significantly fewer particles compared to LCD baselines.

**Weaknesses**

- Limited Task Diversity: The empirical evaluation is restricted to function calling (xLAM)and keyword-constrained generation (CommonGen). The paper would be significantly strengthened by testing on more complex, longer-form structured generation tasks to fully validate the generalizability of the proposed methods.
- Limited Model Scale in Evaluation: The empirical evaluation relies on relatively small base models, specifically Llama 3.1 8B and GPT-2 Large. It is unclear if these expensive proposals are necessary for frontier-class models, whose internal distributions might already strongly align with the required syntax.
- Lack of Direct System Comparisons: While acknowledging concurrent CPU-based parsers like Outlines and XGrammar, the paper fails to provide direct empirical comparisons of throughput  or memory footprint against these established systems.

---

> ### Author Rebuttal · Authors · 2026-03-31
>
> We thank the reviewer for the positive assessment and helpful suggestions.
>
> > **Q1: How do the proposed methods perform on more complex, longer-form structured generation tasks compared to the relatively short xLAM and CommonGen benchmarks?**
>
> We agree that broader evaluation would strengthen the paper. We chose xLAM and CommonGen to cover two settings: (1) structured constraints (function calling/JSON), emphasizing syntax, and (2) keyword-based constraints, emphasizing semantics. The consistent gains indicate the method generalizes beyond a single setting.
>
> We further evaluate on text-to-SQL generation using the Spider benchmark [1], which involves more complex compositional constraints. Following the xLAM setup, we use a random split of 500 prompts for testing and the remainder for distilling a small HMM (hidden size 1024), which is used by P-GCD.
> | Method | Acc (k=1) | Acc (k=4) | Acc (k=8) |
> |--------|----------:|----------:|----------:|
> | LCD | 59.8 | 65.4 | 67.2 |
> | GCD | 59.0 | 66.8 | 69.0 |
> | P-GCD | 56.2 | 67.9 | 69.1 |
>
> In the text-to-SQL setting, (P-)GCD also shows improvement across different number of particles k.
>
> > **Q2: Are the complex P-GCD proposals still necessary and beneficial when applied to larger, frontier-class models, which may already naturally align with the required syntax?**
>
> We agree that larger models better satisfy syntactic constraints, but violations are not fully eliminated, especially in complex structured settings. In practice, systems such as the OpenAI API still rely on constrained decoding frameworks like LLGuidance to ensure correct structured outputs. Existing frameworks (e.g., LLGuidance [2], Outlines, XGrammar) are based on locally constrained decoding (LCD), which does not guarantee constraint satisfaction within a fixed token budget. For agentic systems that depend on reliable function calls, such errors are non-trivial and can be eliminated by GCD.
> P-GCD further improves GCD by incorporating approximate future likelihood. While it adds computation, the overhead is small relative to frontier-model inference and can be parallelized with the base model. Therefore, P-GCD remains beneficial even for stronger models.
>
> > **Q3: Could the authors provide a direct empirical comparison of practical inference throughput and memory footprint against established parsers like Outlines or XGrammar?**
>
> Here we report wall-clock time per token for unconstrained generation, LCD, GCD, and P-GCD using Llama 3 8B on xLAM. Results are averaged over 100 examples (batch size 16); wall-clock is reported as mean ± std over 3 seeds. H denotes the HMM hidden size. All experiments run on a single NVIDIA H100 GPU.
>
>
>
> | Method | ms / token | Slowdown vs unconstrained |
> |---------------------|-------------------|---------------------------|
> | Unconstrained | 1.69 ± 0.01 | 1.00× |
> | LCD | 1.83 ± 0.02 | 1.08× |
> | GCD | 1.86 ± 0.04 | 1.10× |
> | P-GCD (H=1024) | 5.26 ± 0.01 | 3.11× |
> | P-GCD (H=4096) | 8.23 ± 0.63 | 4.87× |
> | P-GCD (H=16384) | 17.30 ± 0.39 | 10.2× |
>
> 1. GCD and LCD add relatively small overhead (<10%) over unconstrained generation.
> 2. P-GCD incurs higher per-token cost than GCD, but for tasks where semantic/probabilistic information about the constraints are important (e.g., CommonGen), it yields faster convergence as an SMC proposal. As shown in Fig. 3, P-GCD reaches ground-truth performance with k=4,
>
> For reference, XGrammar, which is a very optimized CPU-based implementation for LCD, reports \~1.6% - 2.2% overhead on JSON-schema based constrained generation [3], lower than our current GCD implementation (\~10%).
>
> Finally, we emphasize that our current implementation is not system-optimized and we view this as a promising direction for future work: developing optimized, GPU-native implementation for GCD.
>
>
> [1] Yu, T., et al. (2018). Spider: A Large-Scale Human-Labeled Dataset for Complex and Cross-Domain Semantic Parsing and Text-to-SQL Task.
>
> [2] Guidance AI. LLGuidance: Super-fast Structured Outputs. GitHub repository, 2024.
>
> [3] Dong, Y., Ruan, C. F., Cai, Y., Xu, Z., Zhao, Y., Lai, R., and Chen, T. Xgrammar: Flexible and efficient structured generation engine for large language models. Proceedings of Machine Learning and Systems, 7, 2025.

---

> > ### Author Rebuttal · Reviewer_oDaW · 2026-04-03
> >
> > I thank the authors for their feedback. I have no further questions.

---

### Official Review · Reviewer_H8fD · 2026-03-08

**Soundness:** 3
**Presentation:** 3
**Significance:** 3
**Originality:** 3
**Overall Recommendation:** 4
**Confidence:** 2

**Summary:**

The paper introduces (probabilistic) globally constrained decoding, an algorithmic approach for constructing proposal distributions and potential functions within the Sequential Monte Carlo (SMC) sampling framework. Empirical evaluations on the xLAM and CommonGen datasets show that, compared to proposals based on locally constrained decoding, the proposed method achieves substantially better convergence in SMC sampling while requiring far fewer particles.

**Compliance With Llm Reviewing Policy:**

Affirmed.

**Final Justification:**

The additional information for the evaluations during the rebuttal phase make this paper more sound, therefore I maintain my score.

**Key Questions For Authors:**

Q1: Could you evaluate the proposed method on additional base models to further verify its effectiveness? In particular, including several state-of-the-art models would help demonstrate the generality of the approach.

Q2: In Figure 2, could you provide more quantitative descriptions to clarify the results? Specifically, for Figure 2 (right), could you explain more clearly how and to what extent your framework outperforms the baseline method, i.e., LCD?

Q3: In Figure 3, how would you explain that $\phi$-GCD exhibits a non-increasing BLEU score as the number of particles increases?

**Limitations:**

yes

**Strengths And Weaknesses:**

Strengths:

- The paper is well written and easy to follow.

- This paper addresses an important scientific problem.

- The experimental evaluation is rigorous and demonstrates strong performance improvements over the selected baselines.

Weaknesses:

- The evaluations are limited to only one model for each dataset, which restricts the generality of the conclusions.

- The experiments do not clearly quantify the extent to which the proposed method outperforms the selected baselines. More detailed quantitative comparisons would help better illustrate the advantages of the method.

- There's a typo in #382, I guess "As shown in 3" means "As shown in Figure 3".

---

> ### Author Rebuttal · Authors · 2026-03-31
>
> We thank the reviewer for their helpful feedback and suggestions.
>
> > **Q1: Could you evaluate the proposed method on additional base models to further verify its effectiveness? In particular, including several state-of-the-art models would help demonstrate the generality of the approach.**
>
> Thank you for the suggestion. We agree that evaluating on more base models would strengthen the generality of our approach.
>
> To make our evaluation more comprehensive, we further evaluate text-to-SQL generation using the Spider benchmark [1], which involves more complex compositional constraints. Following the setup similar to xLAM, we use a random split of 500 prompts for testing and the remainder for distilling an HMM, which is used by P-GCD.
>
> | Method | Acc (k=1) | Acc (k=4) | Acc (k=8) |
> |--------|----------:|----------:|----------:|
> | LCD | 59.8 | 65.4 | 67.2 |
> | GCD | 59.0 | 66.8 | 69.0 |
> | P-GCD | 56.2 | 67.9 | 69.1 |
>
> In the text-to-SQL setting, (P-)GCD also shows improvement across different numbers of particles k.
>
> Due to the limited rebuttal timeline, we were not able to complete additional large-scale experiments; we are currently running experiments on Qwen, which will be included in the final version.
>
> [1] Yu, T., et al. (2018). Spider: A Large-Scale Human-Labeled Dataset for Complex and Cross-Domain Semantic Parsing and Text-to-SQL Task.
>
> > **Q2: In Figure 2, could you provide more quantitative descriptions to clarify the results? Specifically, for Figure 2 (right), could you explain more clearly how and to what extent your framework outperforms the baseline method, i.e., LCD?**
>
> In the Python-like (Fig. 2, right) setting, P-GCD consistently outperforms LCD by 0.5 - 1.0 accuracy points across all particle counts. Notably, P-GCD with 4 particles achieves comparable performance to LCD with 16 particles, indicating a 4 times improvement in sampling efficiency.
>
> Concretely, Figure 2 shows that:
> 1. GCD vs LCD: The improvement of GCD over LCD comes from enforcing global constraints. In Figure 2 (left), LCD fails to satisfy approximately 5% of constraints (see Table 2), leading to a large performance gap. In Figure 2 (right), the gap is smaller because LCD failure rates are lower (~1%).
>
> 2. P-GCD vs GCD: Across both settings, P-GCD consistently outperforms GCD and LCD by incorporating approximate future likelihood via the HMM, resulting in better proposal quality and convergence.
>
> We will revise Figures 2 and 3 to include explicit quantitative comparisons and clearer descriptions by adding labels showing the precise accuracies. Thank you for your suggestion.
>
>
> > **Q3: In Figure 3, how would you explain that -GCD exhibits a non-increasing BLEU score as the number of particles increases?**
>
> Thank you for raising this important question. We would like to clarify that in SMC sampling, we are not directly optimizing for the BLEU score but instead trying to approximately sample from the ground-truth distribution: as the number of particles increases, the sequences generated from SMC should approach the ground-truth obtained by rejection sampling (dashed horizontal line). Specifically, when using $\phi$-GCD as the potential function, SMC quickly converges with # of particles k >= 8, so the BLEU score trend is non-increasing after convergence. We will make this point clear in the revision.
>
> > **Typo in #382**
>
> Thank you for pointing this out, and we will correct the typo in the revision!

---

> > ### Author Rebuttal · Reviewer_H8fD · 2026-04-03
> >
> > Thank you for the detailed response. The additional information for the evaluations better helps me understand the paper. I prefer to maintain my score since the initial score is suitable enough.

---

### Official Review · Reviewer_sxuZ · 2026-03-15

**Soundness:** 3
**Presentation:** 3
**Significance:** 3
**Originality:** 3
**Overall Recommendation:** 4
**Confidence:** 5

**Summary:**

The paper argues that locally constrained decoding (LCD) is biased because it only masks tokens based on immediate validity, not whether the sequence can still be completed successfully within the remaining token budget, and this makes LCD a weak proposal for SMC. To fix this, the authors introduce globally constrained decoding (GCD), which uses a tensorized finite automaton to check whether each next token can still lead to a valid completion within the horizon, and then extend it to probabilistic GCD (P-GCD) by distilling an HMM that approximates the LM and adds probabilistic lookahead. The result is a stronger SMC proposal/potential that incorporates both hard constraint structure and approximate future likelihood.

**Compliance With Llm Reviewing Policy:**

Affirmed.

**Key Questions For Authors:**

Can you provide a clearer decomposition of the gains from GCD alone versus the additional gains from P-GCD, ideally including runtime or wall-clock comparisons per token/sample to be able to see whether the better convergence offsets the extra HMM machinery in practice?

How sensitive are results to the quality and size of the distilled HMM, does P-GCD remain useful with much smaller/weaker HMMs, and how expensive is distillation relative to downstream decoding savings?

The authors emphasize NFAs over DFAs for compactness, can you say a bit more about the kinds of constraints in xLAM or other agentic workloads where the NFA advantage is decisive in practice, not just in the synthetic separating example?

**Limitations:**

Yes

**Strengths And Weaknesses:**

Strengths:

This paper addresses a real weakness of locally constrained decoding and proposes a technically interesting solution based on tractable proposals for SMC. The core idea gives the paper both a systems contribution and an inference contribution. I think the paper is strongest where it connects these pieces cleanly: FA tensorization, forward/backward masking, the FA–HMM shared circuit structure, and tractable circuit multiplication into a stronger proposal/potential for SMC. Empirically, the results on xLAM and CommonGen are directionally convincing: GCD/P-GCD converge faster than LCD-based proposals, P-GCD improves further over GCD, and unlike LCD they achieve 100% constraint satisfaction in the reported structured function-calling settings. The NFA-vs-DFA case study is also a nice practical point, since it argues the method is not just theoretically broader but potentially much more compact in realistic schema-like settings.

Weaknesses:

My main concern is that the paper’s empirical case is still somewhat narrow relative to the breadth of its claims. The structured-output xLAM setting is a natural fit for FA-based methods, so it is unsurprising that global horizon-aware masking helps there. CommonGen broadens the scope somewhat, but the evaluation there is limited to BLEU and a single HMM-guided setup, which makes it harder to tell how generally useful the proposal construction will be outside these benchmark families. I also wanted more ablations isolating where the gains come from so tensorized global masking alone, the HMM approximation quality, and the use of P-GCD as proposal vs potential are all mixed together, and the practical cost-quality tradeoff is not fully transparent. The method also depends on distilling a reasonably good HMM approximation to the LM, that is plausible, but it adds a nontrivial extra modeling assumption and offline cost that is not deeply interrogated. On the theory side, the paper argues tractability and gives clean constructions, but the approximation quality of the HMM-enhanced proposal to the true constrained LM is mostly validated empirically rather than characterized more sharply.

---

> ### Author Rebuttal · Authors · 2026-03-31
>
> We thank the reviewer for the detailed and insightful feedback.
>
>
> > **W1: xLAM and CommonGen benchmark is somewhat narrow**
>
> We agree that broader evaluation would strengthen the paper, and further evaluate on text-to-SQL generation using the Spider benchmark [1], which involves more complex compositional constraints. Following setup similar to xLAM, we use a random split of 500 prompts for testing and the remainder for distilling an HMM, which is used by P-GCD.
>
> | Method | Acc (k=1) | Acc (k=4) | Acc (k=8) |
> |---|---:|---:|---:|
> | LCD | 59.8 | 65.4 | 67.2 |
> | GCD | 59.0 | 66.8 | 69.0 |
> | P-GCD | 56.2 | 67.9 | 69.1 |
>
> In the text-to-SQL setting, (P-)GCD also shows improvement across different numbers of particles k.
>
> > **Q1. ... gains from GCD alone versus the additional gains from P-GCD, …**
>
> At a high level, GCD targets syntactic correctness of constraints, while P-GCD further captures their semantic aspects, which is implied by our existing experiments:
> 1. xLAM + JSON format: intuitively for structured output tasks, the constraints are primarily syntactic, hence the main improvement here comes from GCD. As shown in Fig. 2(a), the gap between GCD and LCD is much larger than that between P-GCD and GCD.
> 2. CommonGen: the constraints on this dataset are relatively “loose” (compared to xLAM) and the main challenge is semantic coherence, i.e., arranging keywords meaningfully; hence the main improvement (over LCD) here comes from P-GCD instead of GCD.
>
> We will present a more comprehensive discussion in the revision when presenting our empirical results. Thank you for raising this important question.
>
> > **runtime or wall-clock comparisons per token/sample**
>
> For xLAM, we report wall-clock time per token for unconstrained generation, LCD, GCD, and P-GCD using Llama 3 8B. Results are averaged over 100 examples (batch size 16); wall-clock is reported as mean ± std over 3 seeds. H denotes the HMM hidden size. All experiments run on a single NVIDIA H100 GPU.
>
> | Method | ms / token | Slowdown vs unconstrained |
> |---|---|---|
> | Unconstrained | 1.69 ± 0.01 | 1.00× |
> | LCD | 1.83 ± 0.02 | 1.08× |
> | GCD | 1.86 ± 0.04 | 1.10× |
> | P-GCD (H=1024) | 5.26 ± 0.01 | 3.11× |
> | P-GCD (H=4096) | 8.23 ± 0.63 | 4.87× |
> | P-GCD (H=16384) | 17.30 ± 0.39 | 10.2× |
>
> 1. GCD and LCD add minimal overhead (<10%) over unconstrained generation.
> 2. P-GCD incurs higher per-token cost than GCD, but for tasks where semantic/probabilistic information about the constraints are important (e.g., CommonGen), it yields faster convergence as an SMC proposal. As shown in Fig. 3, P-GCD reaches ground-truth performance with k=4, matching GCD/LCD at k=64, reducing number of inference calls to base model by 16 times, which eventually roughly saves 4 times of total compute.
>
>
> > **Q2: How sensitive are results to the quality and size of the distilled HMM ... how expensive is distillation ...**
>
> Thank you for raising this question.We ablate HMM hidden dimension on xLAM (JSON format), reporting the downstream constrained decoding accuracy (Acc) with varying numbers of samples k.
>
> | Hidden size | Acc (k=1) | Acc (k=2) | Acc (k=4) | Acc (k=8) | Acc (k=16) |
> |---|---|---|---|---|---|
> | LCD | 66.4 ± 0.7 | 68.2 ± 0.5 | 69.3 ± 0.4 | 69.7 ± 0.4 | 69.7 ± 0.1 |
> | 1024 | 65.4 ± 0.9 | 68.6 ± 0.7 | 70.2 ± 0.2 | 71.1 ± 0.6 | 71.5 ± 0.1 |
> | 16384 | 66.5 ± 0.8 | 69.2 ± 0.5 | 70.5 ± 0.4 | 71.5 ± 0.1 | 71.6 ± 0.1 |
>
> We find that even relatively small HMMs provide useful guidance yet larger HMMs constantly yield better performance; the advantage is mostly exhibited in low-particle regimes, while the performance gap diminishes as the number of particles increases. This suggests that P-GCD is robust to the quality of the HMM approximation and does not require highly accurate models to be effective.
>
> Distillation Cost:
> HMM distillation consists of two stages: (i) sampling data from the base model, which requires approximately 2 GPU hours on H100 to generate 2M samples (using vLLM), and (ii) training the HMM, which takes approximately 0.2 – 3 GPU hours for hidden sizes ranging from 1024 to 16384. This is a one-time offline cost.
>
> > **Q3: The authors emphasize NFAs over DFAs for compactness … where the NFA advantage is decisive in practice, not just in the synthetic separating example?**
>
> In practical structured generation tasks (e.g., SQL or JSON/function-calling), constraints often involve optional fields and compositional patterns. These lead to state-space explosion when converted to DFAs.
>
> For example, in our text-to-SQL experiments (Spider), for encoding the minimal SQL syntax, NFAs are significantly more compact than their DFA counterparts.
>
> | Representation | NFA size | DFA size |
> |---|---|---|
> | Character-level | 7999 | 57263 |
> | Token-level | 31858 | Compilation OOM (CPU) |
>
> [1] Yu, T., et al. (2018). Spider: A Large-Scale Human-Labeled Dataset for Complex and Cross-Domain Semantic Parsing and Text-to-SQL Task.

---

> > ### Author Rebuttal · Reviewer_sxuZ · 2026-04-05
> >
> > The authors answered my questions, I keep the same score.

---

### Decision · Program_Chairs · 2026-04-30

**Decision:**

Accept (regular)

**Comment:**

This paper proposes globally constrained decoding and its probabilistic extension as improved proposal and potential constructions for SMC-based constrained generation, aiming to provide stronger proposals with guaranteed finite-budget constraint satisfaction and faster convergence to the target constrained distribution.

The reviewers were overall positive. Reviewers agreed that the paper tackles an important problem, introduces a technically elegant framework combining tensorized finite automata, global constraint reasoning, and HMM-based probabilistic guidance, and shows promising empirical gains over LCD-based baselines. The reviewers also raised concerns about the limited breadth of evaluation, restricted model diversity, insufficient clarity on the practical cost-quality tradeoff of P-GCD, and the lack of more direct comparisons to optimized structured-decoding systems.

In the discussion and rebuttal, the authors addressed many of these concerns concretely by adding more experiments, clarifying the roles of GCD and P-GCD, providing wall-clock measurements showing that GCD adds only modest overhead, and including an HMM-size ablation suggesting that even relatively small HMMs are useful. They also strengthened the practical case for NFAs by showing their compactness advantages over DFAs in SQL-style constraints. All reviewers marked their response as fully resolved after the rebuttal.

Given the consistently positive reviewer assessments and the satisfactory response to most of the main concerns, I recommend acceptance. The authors should use the reviewers' suggestions to revise the paper and incorporate promised modifications and clarifications.